# Ibogaine Has Sex-Specific Plasma Bioavailability, Histopathological and Redox/Antioxidant Effects in Rat Liver and Kidneys: A Study on Females

**DOI:** 10.3390/life12010016

**Published:** 2021-12-23

**Authors:** Nikola Tatalović, Teodora Vidonja Uzelac, Milica Mijović, Gordana Koželj, Aleksandra Nikolić-Kokić, Zorana Oreščanin Dušić, Mara Bresjanac, Duško Blagojević

**Affiliations:** 1Department of Physiology, Institute for Biological Research “Siniša Stanković”—National Institute of Republic of Serbia, University of Belgrade, Bulevar Despota Stefana 142, 11060 Belgrade, Serbia; nikola.tatalovic@ibiss.bg.ac.rs (N.T.); teodora.vidonja@ibiss.bg.ac.rs (T.V.U.); san@ibiss.bg.ac.rs (A.N.-K.); zoranaor@ibiss.bg.ac.rs (Z.O.D.); 2Institute of Pathology, Faculty of Medicine, University of Priština, Anri Dinana bb, 38220 Kosovska Mitrovica, Serbia; milica.mijovic@med.pr.ac.rs; 3Institute of Forensic Medicine, Faculty of Medicine, University of Ljubljana, Korytkova 2, 1000 Ljubljana, Slovenia; gordana.kozelj@mf.uni-lj.si; 4Institute of Pathophysiology, Faculty of Medicine, University of Ljubljana, Zaloška 4, 1000 Ljubljana, Slovenia; maja.bresjanac@mf.uni-lj.si

**Keywords:** noribogaine, erythrocytes, liver, kidney, glycogen, redox imbalance, catalase, xanthine oxidase, lipid peroxidation, thiols

## Abstract

Ibogaine induces rapid changes in cellular energetics followed by the elevation of antioxidant activities. As shown earlier in male rats, ibogaine treatment with both 1 and 20 mg/kg b.w. per os led to significant glycogenolytic activity in the liver. In this work, female rats treated with the same doses of ibogaine per os displayed lower liver glycogenolytic activity relative to males, dilatation of the central vein and branches of the portal vein, and increased concentration of thiols 6 h after treatment. These changes were followed by increased catalase activity and lipid peroxidation, and decreased xanthine oxidase activity after 24 h. In kidneys, mild histopathological changes were found in all treated animals, accompanied by a decrease of glutathione reductase (after 6 and 24 h at both doses) and an increase of catalase (6 h) and xanthine oxidase activity (6 and 24 h). Ibogaine did not affect antioxidant enzymes activity in erythrocytes. Bioavailability of ibogaine was two to three times higher in females than males, with similar kinetic profiles. Compared to previous results in males, ibogaine showed sex specific effect at the level of antioxidant cellular system. Effects of ibogaine in rats are sex- and tissue-specific, and also dose- and time-dependent.

## 1. Introduction

Ibogaine is a natural alkaloid from iboga plant (*Tabernanthe iboga* Baill.). It is used in parts of Africa for medical purposes or as a stimulant to overcome fatigue, hunger and thirst as well as in higher doses for spiritual purposes to provoke hallucinations [1,2,3]. In the “ibogaine medical subculture” it is used as anti-addiction agent (against many common addictives, i.e., cocaine, heroin, methadone and alcohol), usually administered in the range from 1–25 mg/kg of body weight (b.w.) as a single oral dose [2,4,5]. However, numerous fatalities associated with ibogaine use have been reported [1,6] revealing a clear need for better understanding of its complex pharmacodynamics and applicable doses. In a study on rats it was shown that ibogaine effects in the central nervous system were sex-specific, i.e., more prominent in females [7]. Ibogaine’s activity is achieved through different types of receptors [1,8,9], but also rapid depletion of ATP, that is followed by the induction of energy-related enzymes, as well as rise of cellular reactive oxygen species (ROS) and antioxidant enzymes activity [10,11,12,13,14,15,16]. Ibogaine itself is not an antioxidant [12], and from experiments in vitro [10,11,12] and ex vivo [14,15,16] it seems that ibogaine is a pro-antioxidative cellular agent. In yeast [11], elevated expression of glycolysis enzymes and CuZn superoxide dismutase (SOD1) protein after 5 h at a dose of 1 mg/L ibogaine (equivalent to peak brain tissue concentration in rats after intraperitoneal (i.p.) application of 20 mg/kg b.w.) were found. Since elevated expression of SOD was not found in brains of rats (dosed with 20 mg/kg b.w., i.p.) after 24 and 72 h [13], results suggest species-, tissue- and time-specific ibogaine effects. This also potentiates the role of adequate dosing as well as the time when effects can be obtained without significant side effects.

Oral application of ibogaine in vivo is followed by its metabolic conversion to noribogaine which is characterized by complex tissue specific pharmacokinetic profiles. Ibogaine in blood is replaced by noribogaine and rapidly cleared [17]. Elimination of noribogaine is slower than ibogaine [1]. At 24 h following per os administration, elimination of ibogaine is about 65% in rats [18], and in humans is even 90% [19], via urine and feces. Biological availability is two to three times higher in female than male rats [18]. Greater bioavailability in female rats was found after i.p. treatment as well [7]. Our previous measurements showed that concentrations of both ibogaine and noribogaine in male rat blood plasma [20] were similar with data obtained on mice [21]. However, comparative data on the pharmacokinetics of ibogaine in male and female rats are not sufficiently detailed and comprehensive [1,7,18] since these publications differ from our study in either strain of rats, whey of application, dose, or time interval. Therefore in this work we measured ibogaine and noribogaine concentrations in the blood plasma of female Wistar rats at our experimental time points and compared them to data obtained in male Wistar rats under the same experimental conditions.

Our previous results obtained in male rats treated in vivo with a single oral dose of ibogaine (1 or 20 mg/kg b.w.), showed significant glycogenolytic activity in hepatocytes 6 and 24 h after administration with no histopathological changes but mild oxidative stress after 24 h in liver [20] as well as changes in antioxidant enzyme activity in kidneys accompanied with moderate morphological changes of proximal tubules, without changes in urinalysis [22]. Since liver glycogenolytic activity is higher in male than female rats and overall hepatic metabolism show significant sex differences [23], ibogaine effects on liver glucose/glycogen metabolism could also be sex-dependent. Moreover, general sex-based differences in pharmacokinetics, pharmacodynamics and adverse drug reactions are well established in literature [24,25]. Along with these facts, numerous reported adverse effects of ibogaine even with fatal outcomes, point to the need to examine sex-based differences in the effects in order to better understand its mechanisms of action because these differences could lead to treatment failure or adverse effects. Here, we examined the effects of ibogaine on erythrocytes, liver and kidneys of female Wistar rats in vivo, 6 and 24 h after treatment with a single oral dose (1 or 20 mg/kg b.w.) in order to compare these results with those obtained in males. Doses and time intervals were chosen according to our previous experiments on male rats [20,22], a suggestion for a maximum oral dosage limit of 1 mg/kg for humans [26], and previous results that explore effects of the dose of 20 mg/kg b.w., i.p. on rat brains [13] as well as studies on yeast exposed to concentration of ibogaine equivalent to peak brain tissue concentration in rats after application of 20 mg/kg b.w., i.p. [11,12]. By histological analysis, we visualized the amount of glycogen in liver (as an expression of energy metabolism), and inspected possible pathological changes in hepatic and renal tissue morphology. Blood glucose concentration, as well as food and water consumption and urine parameters were also monitored. Metabolic function of the liver and kidneys were valued by measuring the activity of xanthine oxidase (XOD; purine turnover) and glutathione S-transferases (GST). Furthermore, possible oxidative damage was estimated by measuring the concentration of TBARS. Protein and non-protein free sulfhydryl (‒SH) groups (thiols) were measured as indicators of thiol-based redox state. General systemic oxidative/antioxidative effect of ibogaine was evaluated by measurement of antioxidative enzymes activity: cytosolic CuZn superoxide dismutase (SOD1), mitochondrial Mn superoxide dismutase (SOD2), catalase (CAT), glutathione peroxidase (GSH-Px) and glutathione reductase (GR).

## 2. Materials and Methods

### 2.1. Animals

Experimental procedures were performed in compliance with the Directive 2010/63/EU on the protection of animals used for experimental and other scientific purposes and were approved by the Ethics Committee for the Use of Laboratory Animals of the Institute for Biological Research “Siniša Stanković”—National Institute of Republic of Serbia, University of Belgrade. Three-month-old healthy female Wistar rats, b.w. 175–250 g, were housed individually at 22 °C, day/night 12 h/12 h, with access to food (rodent laboratory chow made by Veterinarski Zavod Subotica, Serbia) and tap water ad libitum. An experiment was performed on rats in the estrus phase of the estrous cycle, which was determined by examination of a daily vaginal lavage [27].

### 2.2. Chemicals

Ibogaine hydrochloride (PubChem CID: 197059, purity 98.93%) was acquired by Mara Bresjanac (LNPR project funded by the ARRS Program P3-0171).

### 2.3. Experimental Design

All of the experimental procedures as well as doses and time intervals were the same as in our experiments on male Wistar rats that were described in details in our previous papers [20,22]. Ibogaine was dissolved in deionized water (dH_2_O) by strong vortexing (at a stock concentration of 2 mg/mL) and maintained in the dark until use. The animals were randomly divided into five groups (6 animals per group) and treated at 09.00 with one dose of ibogaine per os by gavage (1 or 20 mg/kg b.w.). The control groups were gavaged with an equal amount of dH_2_O. All animals received 1 mL of liquid (dH_2_O or ibogaine solution of appropriate concentration) per 100 g b.w. by gavage. The experimental treatments were: C—control, gavaged with dH_2_O; L6—low dose of ibogaine (1 mg/kg b.w.), sacrificed after 6 h; L24—low dose of ibogaine (1 mg/kg b.w.), sacrificed after 24 h; H6—high dose of ibogaine (20 mg/kg b.w.), sacrificed after 6 h; H24—high dose of ibogaine (20 mg/kg b.w.), sacrificed after 24 h. Control group was initially divided into two subgroups to be decapitated 6 h or 24 h after gavage. Since there were no differences between the two control subgroups they were merged and presented as one control group.

### 2.4. Blood Preparation

Blood was collected immediately after decapitation in tubes coated with heparin (500 I.U. per mL of blood). Plasma and erythrocytes were separated by centrifugation (MiniSpin, Eppendorf, 3000× *g*, 10 min). Plasma was stored at −20 °C for the measurement of concentrations of ibogaine and noribogaine. Erythrocytes were washed three times with saline (0.9% *w/w*) and stored at −20 °C until use. Thawed erythrocytes were hemolyzed by ice-cold dH_2_O and hemolysates were used for measurement of antioxidant enzymes and XOD activities.

### 2.5. Measurement of Concentrations of Ibogaine and Noribogaine in Blood Plasma

A validated liquid chromatography-tandem mass spectrometry (LC-MS/MS) method [20,28] was used for measurement of ibogaine and noribogaine concentrations in blood plasma. The LC-MS/MS instrument included an Agilent 1100 HPLC (Agilent, Santa Clara, CA, USA) system and the tandem quadrupole mass spectrometer Quattro micro™ API (Waters, Milford, MA, USA). Electrospray was operating in positive ionization mode. Data were registered and analyzed using MassLynx 4.1 software. Two SRM transitions were acquired for ibogaine (311→174, 122), noribogaine (297→160, 122) and for the internal standard (325→105, 271). For both ibogaine and noribogaine dissolved in plasma, calibration curves were linear in a range from 0.1 to 100.0 ng/mL. Correlation coefficients were 0.996 or higher. Lower limit of quantitation (LLOQ) was 0.2 ng/mL for both compounds.

### 2.6. Tissue Preparation

Liver was perfused with cold saline (0.9% *w/w*) and excised immediately after decapitation and blood collection. For histological analysis a portion of the left lobe of the liver was used as well as the anterior part of the right kidney (one third of the kidney). Tissue fixation was performed using buffered 4% paraformaldehyde, pH 7.4 for 24 h. The rest of the liver, left kidney and the rest of the right kidney were frozen in liquid nitrogen and stored at −70 °C.

### 2.7. Histological Analysis

After 24 h of tissue fixation, liver and kidney samples were dehydrated through increasing concentrations of ethanol and xylol, and then embedded in Histowax (Histolaboduct AB, Göteborg, Sweden). Thin sections (5 μm) were cut on a HistoCore AUTOCUT microtome (Leica Biosystems, Wetzlar, Germany) and stained by hematoxylin and eosin (H&E) and periodic acid-Schiff (PAS) staining [29]. PAS method is staining glycogen granules in purple-magenta color. Leitz DMRB light microscope (Leica Microskopie and Systeme GmbH, Wetzlar, Germany) with a Leica MC190 HD camera (Leica Microsystems) was used for examination of liver sections while kidney sections were examined using a Leica DM LS2 light microscope (Leica Microsystems, Germany) with a Canon PowerShot S70 camera (Canon U.S.A. Inc., Huntington, NY, USA).

The amounts of glycogen in liver samples were assessed by a quantitative and semi-quantitative analysis as described in our previous paper [20]. Quantitative analysis (determining the number of glycogen-positive cell per 100 cells) was performed at 400× magnifications (objective magnification 40×) also known as the high-power field (HPF) of view. For each sample, five HPFs around the central vein with the strongest glycogen signal were examined and the mean percentage was calculated. Representative micrographs shown in the results section were captured using objective magnification of 20× and camera magnification of 3.2×. Since the glycogen-positive cells can contain a variable amounts of glycogen, we performed semi-quantitative analysis by classifying each sample into one of three categories: + (weak staining), ++ (medium staining) and +++ (strong staining), according to the average intensity of purple-magenta color in all examined fields. Three different examiners interpreted all PAS staining slides. For intensity of PAS staining, prevalent score is given. Fleiss’ kappa coefficient for assessing the inter-rater agreement was 0.479. 

Qualitative H&E analysis of the liver included histomorphological examination of the hepatic lobules for possible pathological changes, for example: hepatocyte edema, dilatation of the central vein, dilatation of portal vein branches, infiltration of lymphocytes and plasma cells in portal spaces, hyperplasia of Kupffer cells, necrosis, and others. Observed dilatation of the central vein and portal vein branches were considered significant if present in more than one third of a sample. Fleiss’ kappa coefficient for assessing the inter-rater agreement was 0.78.

Qualitative H&E analysis of kidney included histomorphological examination of kidney cortex: epithelial cells of renal proximal tubules, shape of the tubular lumens, and morphology of glomeruli, as well as the medulla. Observed morphological changes were classified as: (i) slight changes—regular (round or oval) shape of tubular lumen in at least three quarters of all examined tubules, and rounded nuclei present in all cells but poorly visible in less than one half of all examined tubules; (ii) moderate changes—regular (round or oval) shape of tubular lumen in up to one fifth and stellate in up to three quarters of all examined tubules, as well as rounded and poorly visible nuclei in at least one half of all examined cells and focally-absent in less than one half of all examined tubules, as described in our previous paper [22]. Fleiss’ kappa coefficient for assessing the inter-rater agreement was 0.748.

### 2.8. Measurement of Concentration of TBARS, Thiols and Protein –SH Groups

Perfused and frozen liver and kidney tissue samples were thawed, homogenized (3 × 10 s) and sonicated (3 × 15 s, at 10 kHz) in buffer containing 0.05 M tris(hydroxymethyl)aminomethane (Tris) and 1 mM ethylenediaminetetraacetic acid (EDTA), pH 7.4. Sonicates were than centrifuged (MiniSpin, Eppendorf) for 15 min at 9000× *g* and the supernatants were used for measuring the level of TBARS and total and non-protein free ‒SH groups (thiols). The intensity of lipid peroxidation in liver and kidney were estimated by measuring the concentration of TBARS [30]. Samples were mixed with equal volumes of 0.6% 2-thiobarbituric acid and incubated at 95 °C for 10 min. The absorbance at 532 nm was measured by a Shimadzu UV-160 (Shimadzu Scientific Instruments, Japan). The concentration of TBARS was calculated using malondialdehyde as a standard. The concentrations of total ‒SH groups were measured following the Ellman’s protocol optimized for a microtiter plate [31]. Samples were incubated for 10 min with Ellman’s reagent (3 mM 5,5′-dithiobis-[2-nitrobenzoic acid]) in 0.1 M potassium phosphate buffer (pH 7.3). The absorbance at 412 nm was measured by a Multiskan Spectrum spectrophotometer (Thermo Fisher Scientific Oy, Finland). For determination of thiols concentration, proteins were precipitated by sulfosalicylic acid and supernatant was used for the reaction with Ellman’s reagent. Concentrations of protein –SH groups were calculated by subtracting the concentrations of thiols from concentrations of total –SH groups.

### 2.9. Measurement of the Activities of Antioxidant Enzymes, Glutathione S-Transferases (GST) and Xanthine Oxidase (XOD)

Perfused and frozen liver and kidney tissue samples were thawed, homogenized (3 × 10 s) and sonicated (3 × 15 s, at 10 kHz) in buffer containing: 0.25 M sucrose, 0.05 M Tris and 1 mM EDTA, pH 7.4. Sonicates were than centrifuged (Optima^TM^ L-100 XP Ultracentrifuge, Beckman Coulter, Brea, CA, USA) 90 min at 105,000× *g*, 4 °C, and the supernatants were used for spectrophotometric measurement of enzyme activities using a Shimadzu UV-160 spectrophotometer (Shimadzu Scientific Instruments, Japan). The activities of total SOD and SOD2 were determined by the adrenaline method monitoring the absorbance at 480 nm [32]. SOD activity unit (U) is defined as the amount of enzyme needed for 50% decrease of adrenaline autoxidation rate. SOD2 activity was measured after inhibition of SOD1 by 4 mM KCN. SOD1 activity was calculated by subtracting SOD2 from total SOD activity. For the measurement of SOD1 activity in erythrocyte lysates (erythrocytes does not contain SOD2) hemoglobin was removed according to the method described by Tsuchihashi [33]. CAT activity was measured by monitoring H_2_O_2_ consumption at 230 nm [34]. GSH-Px activity was measured according to modified assay described by Paglia and Valentine [35] by monitoring NADPH consumption at 340 nm. Tert-butyl hydroperoxide was used as a substrate. The activity of GR was measured using the method based on NADPH oxidation and GSSG reduction, monitoring the decrease in absorbance at 340 nm [36]. The activity of GST was measured using 1-chloro-2,4-dinitrobenzene and GSH as substrates and monitoring the increase in absorbance at 340 nm [37]. XOD activity was measured by monitoring uric acid production at 292 nm in the presence of xanthine as a substrate [38]. Enzyme activities were expressed as U per gram of hemoglobin, in erythrocytes, or as U per milligram of protein in liver and kidney.

### 2.10. Measurement of Hemoglobin Concentration

Concentration of hemoglobin in erythrocyte lysates was determined by the method of Drabkin and Austin [39] measuring the absorbance at 545 nm using a Shimadzu UV-160 spectrophotometer (Shimadzu Scientific Instruments, Kyoto, Japan).

### 2.11. Measurement of Protein Concentration

The concentrations of proteins in liver and kidney samples were determined by the method of Lowry measuring the absorbance at 670 nm using a Multiskan Spectrum spectrophotometer (Thermo Fisher Scientific, Vantaa, Finland). Bovine serum albumin was used as a standard [40]. 

### 2.12. Measurement of the Amounts of Consumed Water and Food

Amounts of consumed water and food for each animal were measured at the end of the experiment and expressed as mL of water or g of food per h per kg b.w. The total volume of consumed water was calculated as a sum of the volume applied by gavage and volume that animals have drank.

### 2.13. Measurement of Blood Glucose Concentration

Glucose concentration was measured in the whole blood immediately after decapitation using a Blood Glucose Test Strip TOUCH-IN^®^ Micro (ApexBio, Hsinchu City, Taiwan).

### 2.14. Urine Analysis

To measure the specific gravity, pH value as well as concentrations of glucose, nitrite and ketones in urine, we used Uriscan urine test strips, YD Diagnostics (Yongin, Republic of Korea).

### 2.15. Data Analysis and Statistical Procedures

Statistical analyses of the results were performed following the protocols described by Hinkle, Wiersma and Jurs [41]. Enzyme activities, concentration of TBARS, protein ‒SH groups and thiols were compared by analysis of variance (ANOVA). To test the general effects of ibogaine and the differences between control group and groups treated with ibogaine one-way ANOVA was performed on all five groups (C, L6, L24, H6, H24) followed by Tukey’s HSD post hoc test (all groups compared to each other). The effects of differences between doses of ibogaine and time of exposition were further tested by two-way ANOVA (groups: L6, L24, H6, H24; factors: dose and time). The significance level was *p* < 0.05, and values were presented as the mean ± SEM. The statistical significance of the differences in the intensity of PAS staining and observed histopathological changes were calculated by χ^2^-test followed by the Kolmogorov–Smirnov test (*p* < 0.05 was used as significant). For assessing the inter-rater agreement for histopathology observation (three independent histopathologists performed blinded observations) Fleiss’ kappa coefficient was used. There were no exclusions of animals nor data points.

## 3. Results

### 3.1. Concentration of Ibogaine and Noribogaine in Blood Plasma

Ibogaine after the intake of lower dose was detected in blood plasma after 6 h only in one animal (from 6) being undetectable after 24 h (Table 1 and Table A1). Unlike ibogaine, noribogaine was detected after 6 h in all treated animals, but after 24 h in only one (from 6). Intake of higher dose of ibogaine led to presence of ibogaine in blood plasma of all 6 animals after 6 h, but in only one after 24 h. Noribogaine was detected in all treated animals after both 6 and 24 h. Detected concentration of both ibogaine and noribogaine were higher after higher dose intake, and detected concentrations of noribogaine were higher than ibogaine at all experimental points. Moreover, detected concentration of both ibogaine and noribogaine were higher after 6 h from intake than after 24 h.

### 3.2. Amounts of Consumed Water and Food

There were no significant differences in the amount of consumed water (Table 2). On the other hand 6 h after ibogaine application rats had consumed less food (both one-way ANOVA, F = 4.81, *p* < 0.01 as well as two-way ANOVA shoved significant differences with time as significant factor F = 18.75, *p* < 0.001; results of Tukey’s HSD test are shown in Table 2). These differences in food consumption are at least in part an artefact of experimental design and cannot be uncritically considered solely as an ibogaine effect. For a detailed explanation see Table A2.

### 3.3. Blood Glucose

Six hours after treatment, blood glucose level was significantly higher in rats treated with 20 mg/kg compared to those treated with 1 mg/kg (one-way ANOVA, F = 2.80, *p* < 0.05, post hoc Tukey’s *t*-test *p* < 0.05), but without significance between them and control group. No other statistically significant differences were found (Table 2).

### 3.4. Urine Analysis

Urine analysis showed no presence of glucose or nitrite in either the control group or any treated group. Concentration of ketones as well as the pH value and urine specific gravity were not different in the treated groups compared to control (data not shown).

### 3.5. Presence of Glycogen and Color Intensity of Periodic Acid-Schiff (PAS) Staining in the Liver

While glycogen positive cells appeared to display glycogenolytic activity in ibogaine treated rats, no statistically significant differences from controls were found (F = 1.71). Concentration of glycogen seems to fall in hepatocytes after ibogaine intake, but χ^2^-test showed no significant differences (Table 3a, Figure 1a).

### 3.6. Histopathological Analysis of Liver and Kidneys 

Statistical analyses showed that ibogaine treatment provoked morphological changes in liver and that the number of changes was significantly different between groups (χ^2^ = 9.58, *p* < 0.05) (Table 3b and Figure 1b). Morphological changes included dilatations of central vein and portal vein branches. More changes were found 6 h after low dose intake than after 24 h; higher dose led to dilatation of central vein and portal vein branches after 6 h, being more pronounced than after 24 h.

Ibogaine treatment led to slight or moderate pathological changes at the level of proximal tubules and tubular epithelial cells in kidney (χ^2^ = 16.67, *p* < 0.01). Histopathological changes were more pronounced after high dose (*p* < 0.01; more moderate changes were observed) (Table 3c and Figure 1c). There were no significant differences between 6 and 24 h group for neither of two doses. 

### 3.7. Concentration of TBARS, Thiols and Protein ‒SH Groups and Enzyme Activities

There were no changes in the activity of antioxidant enzymes in erythrocytes of rats treated with 1 and 20 mg/kg of ibogaine after 6 and 24 h (Figure 2a; no significant ANOVA).

However, treatment with lower dose of ibogaine elevated CAT (one-way ANOVA, *p* < 0.01) and decreased XOD (*p* < 0.05) activities in liver after 24 h compared to non-treated controls (Figure 2b). Higher dose of ibogaine elevated the concentration of TBARS (*p* < 0.05) after 24 h, but Tukey’s HSD post hoc test showed significance only compared to both lower doses. Both experimental ibogaine doses led to elevated cellular thiols (one-way ANOVA, *p* < 0.001; Tukey’s HSD post hoc test *p* < 0.01 for higher dose) after 6 h, that were at the level of controls after 24 h. Two-way ANOVA analyses confirmed that generally ibogaine at a dose of 20 mg/kg elevated the amount of TBARS compared to a dose of 6 mg/kg. Moreover, two-way ANOVA showed that CAT activity after 24 h after ibogaine treatment was higher comparing to levels after 6 h. On the other hand, XOD activity and thiols were lower after 24 h compared to levels after 6 h from treatment. 

In kidneys, lower dose of ibogaine had no effect on antioxidant enzymes level after 6 h, but GR activity decreased while XOD activity increased after 24 h (one-way ANOVA, *p* < 0.05, both enzymes) compared to controls (Figure 2c). Higher dose elevated CAT (*p* < 0.05) and XOD (*p* < 0.05) activities and deceased GR (*p* < 0.05) activity after 6 h relative to control non-treated animals. Elevated XOD and decreased GR levels persisted up to 24 h (significant differences relative to controls *p* < 0.001 and *p* < 0.01, respectively). Measurement of thiols showed that only higher dose had significant effect relative to all other groups except control (one-way ANOVA, significant effect, *p* < 0.01). When two-way ANOVA was applied considering differences between the effects of dose and time significant time effect was calculated for CAT activity (levels after 6 h were higher than after 24) and dose effect for GR (higher dose had significantly higher effect than lower dose on GR activity) and XOD (higher dose also had significantly higher effect on XOD activity comparing to lower dose). Two-way ANOVA showed significant difference in the presence of thiols after 6 h (two-way ANOVA time effect (*p* < 0.01), but this is more expressed when animals received a higher dose (two way ANOVA, interaction, *p* < 0.05).

## 4. Discussion

Blood plasma concentrations of ibogaine and noribogaine measured in this experiment were generally in accordance with literature data. An earlier publication showed that there was no significant difference in ibogaine kinetic parameters between males and females [18]. Peak plasma concentrations, areas under concentration-time curve and oral bioavailability were greater in females compared to males [1,7,18], but the absorption of oral suspension in rats was generally variable and incomplete [1]. In our experiment, proportions of females in which blood plasma ibogaine and noribogaine were detected in each experimental group were very similar to those in males [20], suggesting similar pharmacokinetics. On the other hand, concentrations of ibogaine and noribogaine in blood plasma were 2–3 times greater in females compared to males [20], Table A1 signifying sex difference in ibogaine and noribogaine oral bioavailability. Furthermore, noribogaine concentrations were several times greater than ibogaine both 6 and 24 h after ingestion. It seems that ibogaine acts quickly [10,11], and our results after 6 and 24 h may illustrate the effects of rapid ibogaine action, or be noribogaine-mediated. Therefore, some indirect and/or receptor mediated noribogaine in vivo effects are possible additional factors that could be taken into account when ibogaine action is considered.

Ibogaine induces different changes in cellular energetics, and our previous results in males showed significant glycogenolytic activity that was connected with ibogaine influence on glucose metabolism [20]. In both males and females, food consumption six hours after treatment with either dose was smaller compared to 24 h groups. This could be due to the stress, the circadian rhythm i.e., rat alimentary habits, transient hunger suppression by water i.e., ibogaine solution ingested by gavage, and possibly fast clearance of ibogaine and metabolic, gastrointestinal and/or behavioral (appetite) effects of ibogaine. However, the difference in food consumption was not clearly correlated with glucose levels in blood after 6 h as well as glycogenolytic activity. There are significant individual differences in liver glycogen content (according to PAS staining) within groups as well as in food consumption. Lower ibogaine dose after 6 h led to small decrease in blood glucose, suggesting higher uptake of glucose from blood, but higher dose led to an elevation, whereby it seems that glucose was mobilized into blood, but without significant effect on the liver glycogen level. This implies that ibogaine effects on metabolism are relatively fast and occurs early. Our results presented here show that in females, ibogaine glycogenolytic potential is also expressed, but to a lower extent than in males. High glycogenolytic activity in liver was expressed in six out of ten of examined animals after 6 h from treatment, and five out of nine after 24 h suggesting also individual variability of ibogaine glycogenolytic effects. Although the number of glycogen positive cells appeared to display glycogenolytic activity in ibogaine treated rats, no statistically significant differences from controls were found. Mean proportions of glycogenolysis displaying cells are in accordance with our results found in males [20], suggesting that ibogaine induces some glycogenolysis also in females. Metabolic activity and carbohydrate utilization in liver is lower in females and, thus, under different hormonal control according to different body composition and physiological demands [23]. Our results are in accordance with observed sex differences that can influence ibogaine effects. Taken together, it seems that there are changes in glucose metabolism to an extent that is not statistically proved when compared to controls, but when differences were estimated among treated animals. Together the findings suggest that ibogaine consumption induced changes in glucose metabolism, but glucose stores and homeostatic capacity was not significantly affected. Our results do not exclude the possibility that more significant changes on the level of glucose metabolism occurs earlier, that cannot be detected after 6 and 24 h. 

Ibogaine dose-dependently disturbs ROS homeostasis [11,12]. Our results show that there were no changes in the activity of antioxidant enzymes in erythrocytes after ibogaine at either experimental dose after 6 and 24 h. No changes of erythrocytes antioxidant enzymes activity were also recorded in males [20]. Our previous results [10] showed elevation of SOD2 and GR activity in erythrocytes followed by ATP release after 1 h of whole blood incubation with 10 and 20 μM ibogaine in vitro. However, concentration of ibogaine in blood plasma of female rats 6 h after per os administration were mostly below detection limit (0.2 ng/L i.e., 0.644 nM) for lower ibogaine dose (1 mg/kg b.w.) and 7.6 ng/L (i.e., 24.5 nM) on average for higher dose (20 mg/kg b.w.). These concentrations of ibogaine were likely too low to provoke significant effects on antioxidant defense in erythrocytes or the release of ATP. Kubiliene et al. [21] showed that the maximal concentration of both ibogaine and its metabolic product noribogaine in mice blood was achieved after just 30 min. Lack of response in erythrocyte antioxidant enzyme activities after 6 h could also be due to rapid ibogaine metabolism, and its earlier and transient effectiveness. There are reports that the difference between ibogaine measured in whole blood and in plasma lays in thrombocytes suggests that ibogaine is being sequestrated by thrombocytes instead of erythrocytes [42]. However, there are no data about concentration of either ibogaine or noribogaine in erythrocytes as well as the effect of noribogaine on erythrocytes antioxidant enzymes.

In the liver, according to our result, the first line of antioxidant action after ibogaine consumption manifests as an elevation of cellular thiols. This can be explained also as redox regulation which includes both glutathione and other small ‒SH containing molecules including metabolic cofactors such as coenzyme A (involved in biosynthesis and degradation of fatty acids). Depletion of ATP shifts the cellular redox state toward more oxidative, and enhanced synthesis of non-protein ‒SH containing molecules seems to be an adaptive response. However, since synthesis of GSH requires energy i.e., reductive agents (ATP, NADPH) it seems that elevation of thiols comes not from enhanced GSH synthesis but from other classes of thiols and, thus, metabolic sources such as the aforementioned coenzyme A, but also glutaredoxin and peroxiredoxin. Since there is no difference in the activity of GSH-depending enzymes (GPx, GR and GST) it seems that glutathione turnover, based on the activity of these enzymes, is not altered. 

Although there are data showing rapid ibogaine effects in vitro, the bulk of antioxidant response in vivo seems to come later. CAT activity increases 24 h post ibogaine suggesting intensive hydrogen peroxide-mediated metabolism, long after first ibogaine and noribogaine cellular and pharmacological action. Furthermore, the level of XOD activity at the same time is lower, suggesting low catabolic metabolism of purine nucleotides. In males, XOD activity was elevated by ibogaine, which may indicate more intensive catabolic processes in male rat liver, along with elevation of lipid peroxides concentration [20]. The differences found in female rat metabolism post ibogaine point to other energetic metabolic processes that could be involved in females. Previous results showed that elevated protein expression of SOD takes place after 24 h in the male rat brain and after 6 h in yeast [11,13]. It seems that ibogaine provokes different energetic metabolic processes to restore ATP reserves, followed by antioxidant enzyme activity. Since we found no significant change in glycogen reserves in our female rats after per oral ibogaine it seems that intensive glycogen re-synthesis is not achieved. On the other hand, ibogaine is sequestered in adipose tissue [43], and could also deplete ATP reserves there, which could be followed by fatty acid release into the blood stream. Fatty acids can be used for liver ATP restoration, via β-oxidation in peroxisomes followed by hydrogen peroxide production [44]. Thus, elevation of CAT activity (mainly located in peroxisomes) can be attributed to this process. Intensive metabolic activity after 24 h led to elevation of lipid peroxidation, but only for a higher dose of ibogaine. However, histopathological examination showed dilatation of the central vein and branches of the portal vein after ibogaine treatment at both doses and time. This was not the case in males. Observed vasodilation could be induced not only by ibogaine or noribogaine, but indirectly by different metabolics that locally affect blood vessels. The findings may imply either higher liver sensitivity to ibogaine or its greater bioavailability in females relative to males. It has been shown that vasoconstrictor tone is lower and endothelium-dependent vasodilation is higher in females that reflects an interplay between circulating sex steroid hormones, their receptors, and sex steroid-independent mechanisms [45,46]. However, no other histopathological changes were observed (hepatocyte edema, infiltration of lymphocytes and plasma cells in portal spaces, hyperplasia of Kupffer cells, necrosis) that led us to conclude that overall histopathological changes are only slight to mild. Since the results obtained here in female rats differ from those observed in males [20], this speaks in favor of sex differences in molecular and physiological mechanisms of ibogaine action. It seems that in males, dominant ibogaine stimulated metabolic process in the liver is glycogen mediated, which is not the case in females, where a lipid-mediated process may play a major role. 

In kidneys, both doses of ibogaine led to a decrease of GR activity. This could be a consequence of lesser availability of NADPH that persisted for 24 h. A similar result was found in males [22] suggesting similar metabolic effects of ibogaine and processes involved in its action in the kidney. Elevated CAT activity in kidneys 6 h post ibogaine suggests increased production of hydrogen peroxide. Intensification of metabolic processes is suggested also by the elevation of XOD activity that points to purinergic catabolism and elevated degradation of adenine that is not so pronounced in the liver. This elevation is greater at the higher dose of ibogaine. Taken altogether it seems that kidney is faced with significant oxidative pressure induced by ibogaine after 6 h, particularly at the higher dose of ibogaine where additional elevation of thiols was also recorded. This could include glutathione, but also other small ‒SH group(s) containing molecules from cysteine and coenzyme A to glutaredoxin and peroxiredoxin. Elevation of thiols can serve as metabolic support (coenzyme A) or an enhancement of antioxidant defense (glutaredoxin and peroxiredoxin). After 24 h, XOD activity was still higher than in controls, and GR activity was still lower than in controls suggesting that catabolic processes were still ongoing. Similar GR activity decrease found in kidneys of females and males after ibogaine, suggests a common fall in the availability of NADPH and slower GSH turnover. These metabolic changes were followed by slight to moderate histopathological changes that were more pronounced in female rats after a higher dose of ibogaine. This suggests a faster, intensive metabolism provoked by ibogaine that resulted in disturbed redox homeostasis, and ROS-mediated histopathological changes. Parenchymal degeneration is usually a manifestation of toxic damage that can be caused by drugs and medications. Ibogaine could affect epithelia of proximal tubules that are the most sensitive to different damaging agents which can involve oxidative stress. Anyway, ibogaine treatment did not provoke changes in urine-specific gravity and pH or glucose, nitrite and ketones concentrations that suggest that kidney function was not impaired. Similar changes were found in males [22], but a higher dose provoked mostly slight morphological changes of proximal tubules after 6 h, suggesting somewhat higher sensitivity of females to ibogaine. 

Our previous results did not show any serious toxic and harmful ibogaine effects on male rats treated with per os doses that did affect energetic metabolism and redox balance [20,22]. Here, we had been trying to gain more insight into the mechanisms of its action in vivo on female rats that could contribute to its safer use. Currently there are three ongoing clinical trials that investigate ibogaine treatment of addiction (EU Clinical Trials Register 2014-000354-11; ClinicalTrials.gov Identifier: NCT03380728; ClinicalTrials.gov Identifier: NCT04003948). Our results suggest the need to carry out preclinical studies in both sexes since important sex-based differences in ibogaine and noribogaine plasma concentrations have been shown as well as sex-based differences in metabolic effects, which could potentially lead to the therapy’s failure and/or adverse effects. Additionally, since ibogaine clearance was shown to be very fast, the results of this study might also suggest that some of effects did happened before our first time point (6 h) pointing in the direction of further research, a shorter period after ibogaine application. Furthermore, thorough investigation of ibogaine/noribogaine tissue concentration would be helpful for the better understanding of its effects. Studies of metabolic, redox/antioxidant and pathological effects on other cells/tissues of interest are needed as well, especially the heart due to its susceptibility to the adverse effects of ibogaine. Due to evident sex differences, more comparative sex-dependent studies on different organisms and levels are needed.

## 5. Conclusions

Our results show that ibogaine and noribogaine plasma bioavailability was greater in females than males. Ibogaine induced some glycogenolytic activity and mild histopathological changes in the liver of a few female rats. In kidneys, ibogaine induced slight to moderate histopathological changes. Treatment with ibogaine was accompanied with redox imbalance in both liver and kidneys. At the level of antioxidant enzymes, ibogaine effects are tissue-specific, as well as dose- and time-dependent. Additionally, ibogaine effects were shown to be sex specific.

## Figures and Tables

**Figure 1 life-12-00016-f001:**
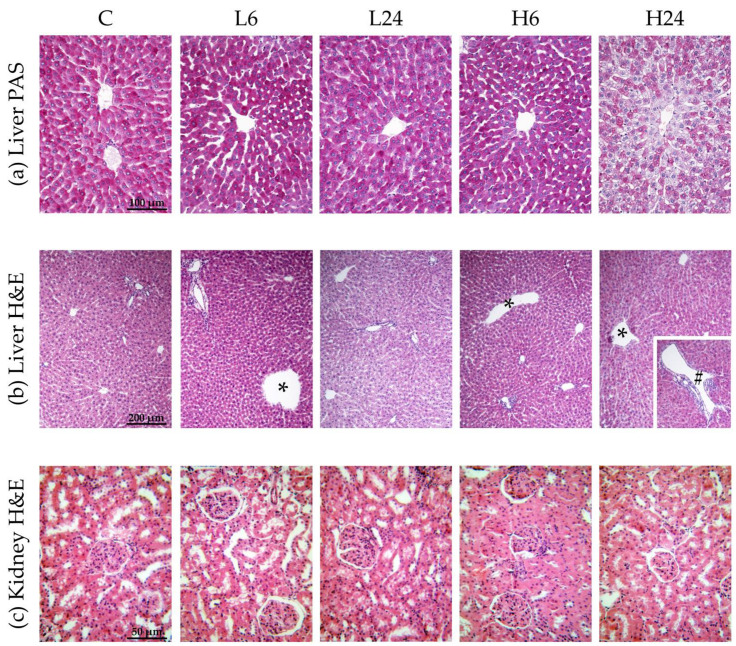
Micrographs of female rats’ liver and kidney after treatment with ibogaine. The control group (C) was treated with dH_2_O; other groups were treated with ibogaine as follow: 1 mg/kg b.w. decapitated after 6 h (L6), 1 mg/kg b.w. decapitated after 24 h (L24), 20 mg/kg b.w. decapitated after 6 h (H6), 20 mg/kg b.w. decapitated after 24 h (H24). (**a**) Liver, PAS staining, objective magnification 20x, glycogen granules in the cytoplasm are colored purple-magenta. (**b**) Liver, H&E staining, objective magnification 10×, showing: C—regular hepatic tissue structure, L6—dilatation of central vein (*), L24—regular hepatic tissue structure, H6—dilatation of central vein (*), H24—dilatation of both central vein (*) and smaller branch of portal vein (#). (**c**) Kidney H&E staining, objective magnification 40×, showing: C—regular renal tissue structure, L6—slight pathological changes at the level of proximal tubules and tubular epithelial cells, L24—slight pathological changes, H6—moderate pathological changes, H24—moderate pathological changes.

**Figure 2 life-12-00016-f002:**
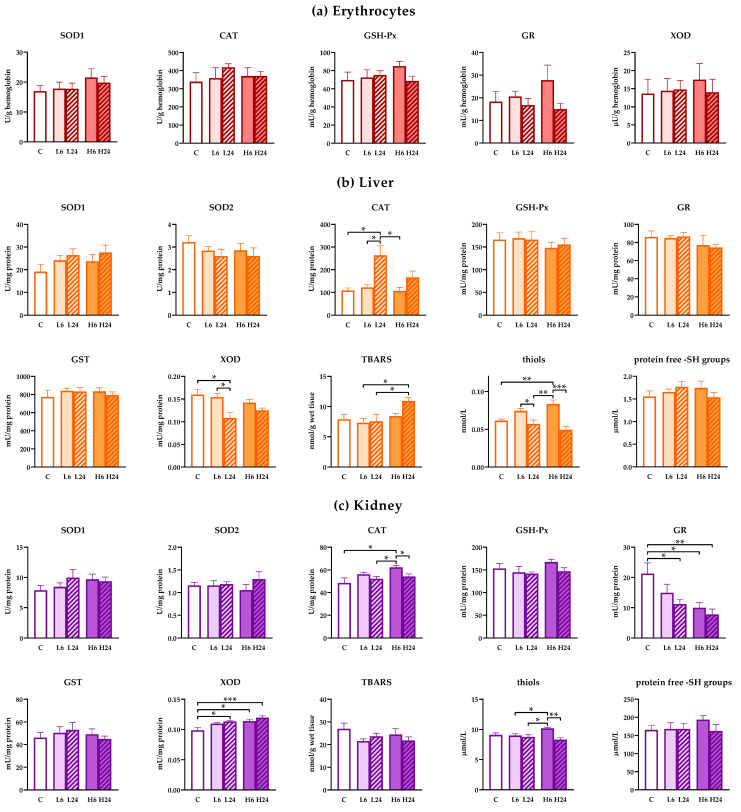
Antioxidant enzyme, glutathione S-transferases (GST) and xanthine oxidase (XOD) activities and concentration of TBARS, thiols and protein ‒SH groups in erythrocytes, liver and kidney of female rats treated with ibogaine. (**a**) Ibogaine caused no significant changes in the activities of either superoxide dismutase (SOD), catalase (CAT), glutathione peroxidase (GSH-Px), glutathione reductase (GR) or XOD in erythrocytes. (**b**) In liver ibogaine altered the activities of CAT (one-way analysis of variance (ANOVA): F = 6.77, *p* < 0.001; two-way ANOVA—time: F = 12.26, *p* < 0.01) and XOD (one-way ANOVA: F = 3.99, *p* < 0.05; two-way ANOVA—time: F = 9.16, *p* < 0.01) as well as concentrations of TBARS (one-way ANOVA: F = 2.99, *p* < 0.05; two-way ANOVA—time: F = 8.02, *p* < 0.05) and thiols (one-way ANOVA: F = 11.05, *p* < 0.001; two-way ANOVA—time: F = 29.67, *p* < 0.001). (**c**) In kidney ibogaine altered the activities of CAT (one-way ANOVA: F = 3.63, *p* < 0.05; two-way ANOVA—time: F = 9.02, *p* < 0.01), GR (one-way ANOVA: F = 4.80, *p* < 0.01; two-way ANOVA—dose: F = 4.23, *p* < 0.05) and XOD (one-way ANOVA: F = 6.63, *p* < 0.001; two-way ANOVA—dose: F = 4.86, *p* < 0.05) as well as the concentrations of thiols (one-way ANOVA: F = 5.10, *p* < 0.01; two-way ANOVA—time: F = 12.42, *p* < 0.01; interaction: F = 7.49, *p* < 0.05). The control group (C) was treated with dH_2_O; other groups were treated with ibogaine as follow: 1 mg/kg b.w. decapitated after 6 h (L6), 1 mg/kg b.w. decapitated after 24 h (L24), 20 mg/kg b.w. decapitated after 6 h (H6), 20 mg/kg b.w. decapitated after 24 h (H24). Significant differences between individual groups obtained by Tukey’s HSD test are shown in the figure; *—*p* < 0.05, **—*p* < 0.01, ***—*p* < 0.001.

**Table 1 life-12-00016-t001:** Concentrations of ibogaine and noribogaine in blood plasma (mg/L).

		C	L6	L24	H6	H24
Ibogaine	No. detected/total no.	0/6	1/6	0/6	6/6	1/6
	Mean	n.a	0.0026	n.a	0.0076	0.0008
	SEM	n.a	n.a	n.a	0.0010	n.a
Noribogaine	No. detected/total no.	0/6	6/6	1/6	6/6	6/6
	Mean	n.a	0.0128	0.0112	0.1602	0.0086
	SEM	n.a	0.0009	n.a.	0.0047	0.0020

Lower limit of quantitation (LLOQ) was 0.2 ng/mL for both ibogaine and noribogaine; means and standard errors of the mean (SEM) were calculated taken into account only animals were ibogaine/noribogaine were quantified. The control group (C) was treated with dH_2_O; other groups were treated with ibogaine as follow: 1 mg/kg b.w. decapitated after 6 h (L6), 1 mg/kg b.w. decapitated after 24 h (L24), 20 mg/kg b.w. decapitated after 6 h (H6), 20 mg/kg b.w. decapitated after 24 h (H24). Concentrations of ibogaine and noribogaine in blood plasma of each treated rat are shown in Table A1. n.a.—not applicable.

**Table 2 life-12-00016-t002:** Amounts of consumed water and food and blood glucose concentrations.

		C	L6	L24	H6	H24
Water [mL/h/kg b.w.]	Mean	3.877	4.953	5.549	4.086	4.342
	SEM	0.301	0.464	0.530	0.693	0.647
Food [g/h/kg b.w.]	Mean	2.303	1.278 ^a^	2.728 ^b^	0.757 ^a^	2.469 ^b^
	SEM	0.467	0.483	0.202	0.377	0.316
Blood glucose [mmol/L]	Mean	6.830	5.650 ^a^	6.408	7.100 ^b^	6.842
	SEM	0.418	0.180	0.377	0.392	0.325

The control group (C) was treated with dH_2_O; other groups were treated with ibogaine as follow: 1 mg/kg b.w. decapitated after 6 h (L6), 1 mg/kg b.w. decapitated after 24 h (L24), 20 mg/kg b.w. decapitated after 6 h (H6), 20 mg/kg b.w. decapitated after 24 h (H24). Different letters in superscript show statistically significant difference (a is different from b) at the level of *p* < 0.05 (Tukey’s honestly significant difference (HSD) test). For detailed explanations on food consumption see Table A2.

**Table 3 life-12-00016-t003:** Presence of glycogen expressed as the number of cells that contain glycogen per 100 cells total (% of glycogen positive cells) and color intensity of periodic acid-Schiff (PAS) staining in the liver (a) and histopathological analysis of liver (b) and kidneys (c) stained by hematoxylin and eosin (H&E) staining.

**(a) Liver PAS Staining**						
		**C**	**L6**	**L24**	**H6**	**H24**
% of glycogen positive cells	Mean	91.83	57.75	59.00	66.50	77.00
	SEM	3.18	15.75	10.72	12.81	6.74
Intensity of PAS staining	+++	6/6	2/4	1/4	4/6	3/5
	++	0/6	0/4	3/4	0/6	2/5
	+	0/6	2/4	0/4	2/6	0/5
**(b) Liver H&E Staining**						
		**C**	**L6**	**L24**	**H6**	**H24**
No morphological changes		6/6	1/4	3/4	3/6	1/5
Dilatation of central vain and portal vain branches		0/6	3/4	1/4	3/6	4/5
**(c) Kidney H&E Staining**						
		**C**	**L6**	**L24**	**H6**	**H24**
No morphological changes		6/6	0/4	0/4	0/6	0/6
Slight changes		0/6	3/4	3/4	0/6	1/6
Moderate changes		0/6	1/4	1/4	6/6	5/6

(a) Percentage (%) of glycogen positive cells are expressed as the mean and standard error of the mean (SEM). Intensity of PAS staining was determined by classifying each sample in one of the three classes: + (weak staining), ++ (medium staining) and +++ (strong staining). (b) Morphological changes in liver are expressed as the number of animals with dilatation of central vein and smaller portal vein branches per total. (c) Morphological changes in kidney are expressed as the number of animals with slight or moderate pathological changes at the level of proximal tubules and tubular epithelial cells per total. The control group (C) was treated with dH_2_O; other groups were treated with ibogaine as follow: 1 mg/kg b.w. decapitated after 6 h (L6), 1 mg/kg b.w. decapitated after 24 h (L24), 20 mg/kg b.w. decapitated after 6 h (H6), 20 mg/kg b.w. decapitated after 24 h (H24).

## Data Availability

The data used to support the findings of this study are included within the article.

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
