# Peer review of "Ibogaine Has Sex-Specific Plasma Bioavailability, Histopathological and Redox/Antioxidant Effects in Rat Liver and Kidneys: A Study on Females"

_life, 2021, doi:10.3390/life12010016_

Round 1
Reviewer 1 Report
The article by Tatalovic et al study the effect of ibogaine in female rats using a number of parameters from levels in blood, to effect on food, glucose, glycogen and anti-oxidant system
The study is interesting but needs substantial improvement overall (from title to introduction, discussion).
Specific comments.
- the title is a bit misleading and not representative of the study results. The study does not have a focus the effect of ibogaine on antioxidant system but it is more like a toxicological study. Despite the fact that most of the discussion is on the antioxidant system, results focus on other parameters as well. (effect on food, glucose, vessel dilatation etc etc). Title can be improved.
- Lines 55-57. How Central nervous system affects bioavailability?
- Lines 59-62. In this study, PK are not really performed on Female mice. Which studies are the authors referring when they say that data are not detailed and comprehensive? The should be detailed here.
- Introduction needs improvement as it talks about the results/methodology of the present study in a lot of detail.
- Line 96. Specify the directive number
- Line 293. since food consumption was less in 6 hours after H and L it cannot be by chance. Further since at H6 food was less than in L6 this is telling us that ibogaine has some effect on the appetite/behavior of the animals. Perhaps stress etc, coupled to the fast clearance of ibogaine.
- Lines 395-399. Based on these lines, why it this study different in terms of PK as detailed in lines 59-61?
- Lines 407-410. This does not seem to have a flow with the aforementioned text. There was not precedent on ROS to support these sentences
- Line 516-517. this conclusion cannot be supported from the present study
- While a lot of data were presented the discussion focuses mostly on the antioxidant system. Some results are not even mentioned. If not needed, perhaps should not be included in this study to begin with and include only results relevant to the title and the main aim of the study.
Author Response
Dear Reviewer,
Thank you for your comprehensive review and very useful comments. We tried to improve our article according to your suggestions. Some sentences in Introduction have been changed, Discussion has been extended and Conclusions and Title modified.
Replies to specific comments:
- the title is a bit misleading and not representative of the study results. The study does not have a focus the effect of ibogaine on antioxidant system but it is more like a toxicological study. Despite the fact that most of the discussion is on the antioxidant system, results focus on other parameters as well. (effect on food, glucose, vessel dilatation etc etc). Title can be improved.
The title is changed according to your suggestions.
- Lines 55-57. How Central nervous system affects bioavailability?
Thank you. Our sentence had linguistic error and original meaning was not expressed. We corrected the sentence.
- Lines 59-62. In this study, PK are not really performed on Female mice. Which studies are the authors referring when they say that data are not detailed and comprehensive? The should be detailed here.
In revised manuscript we emphasized our claim was related to female rats. Our literature search yielded 3 publications dealing with pharmacokinetic properties of ibogaine in female rats (actually, comparing the pharmacokinetic properties of ibogaine in male and female rats):
- Abstract from National Institute on Drug Abuse 55th Annual Scientific Meeting: The College on Problems of Drug Dependence, Inc. Volume II NIDA Research Monograph Series by Jeffcoat et al. (referenced in the manuscript). Male and female Sprague-Dawley rats (4 per group) were treated per os with 5 or 50 mg/kg of ibogaine, or intraperitoneally with 5 mg/kg. Ten blood samples were taken during 24 hours (exact time points not specified). Concentrations of ibogaine and noribogaine were measured.
- A. Upton, Presented at the NIDA Ibogaine Review Meeting, Rockville, MD, 1995. This report is briefly reviewed in Alper, K. 2001 (referenced in the manuscript). It says “As in the study cited above by Jeffcoat (153), peak levels and bioavailability were greater in female than in male rats.”
- Pearl, S.M.; Hough, L.B.; Boyd, D.L.; Glick, S.D. Sex Differences in Ibogaine Antagonism of Morphine-induced Locomotor Activity and in Ibogaine Brain Levels and Metabolism. Pharmacol. Biochem. Behav. 1997, 57, 809–815. DOI: 10.1016/s0091-3057(96)00383-8 (referenced in the manuscript). Male and female Sprague-Dawley rats were treated with ibogaine intraperitoneally 40 mg/kg. Concentrations of ibogaine and noribogaine in plasma were measured 1, 5 and 24 hours after the treatment.
Our experiment was performed on Wistar rats treated per os with ibogaine; doses were 1 or 20 mg/kg; time of sacrificing 6 and 24 hours after the treatment. Based on the available literature we couldn’t make an accurate estimation of the concentrations of ibogaine and noribogaine in the plasma of rats in our experimental time points; therefore, we decided to measure it. Moreover, it helped us to better understand ibogaine effects in rats (in our experiment females) as well as to make a better comparison with our published results on males.
- Introduction needs improvement as it talks about the results/methodology of the present study in a lot of detail.
It seems that the previous results and methodology of the present study are specified in introduction in a lot of detail. But, one of the problems with potential ibogaine application is adequate dosing and appropriate benefits without side effects that can lead to mortality. Therefore, we tried to explain not just experimental approach but the problem we tried to solve. To highlight this, we added the sentence: “This potentiates the role of adequate dosing as well as the time when effects can be obtained without significant side effects.” Oxidative stress is one of possibilities although ibogaine is considered as pro-antioxidant. Some of our previous publications deal with that topic and show that ibogaine strongly influence redox homeostasis and antioxidant enzymes. We are on the track with the latest knowledge about this subject. However, we shortened some sentences and edited original introduction thus it doesn’t contain too much methodological details. We added a reference that showed sex-based differences in liver glycogenolytic activity that underlying the need of the examination of the possibility that ibogaine can express sex specific effects on liver metabolism. We also expanded Introduction with reference on general sex-based differences in pharmacokinetics, pharmacodynamics and adverse drug reactions. This emphasizes the importance of appropriate dosing and thus the need for investigation of ibogaine (potentially) sex specific effects.
- Line 96. Specify the directive number
The number is added.
- Line 293. since food consumption was less in 6 hours after H and L it cannot be by chance. Further since at H6 food was less than in L6 this is telling us that ibogaine has some effect on the appetite/behavior of the animals. Perhaps stress etc, coupled to the fast clearance of ibogaine.
Thanks for suggestion, we agree with your comment. We have extended Discussion with commentaries of these facts. The sentence with explanations was originally submitted in appendix. We moved it into Discussion. However, food consumption in H6 was not less than in L6. Letters in superscript are the same (L6a and H6a) which represent that post hoc test did not show significant difference between H6 and L6. Although there is a large difference in means between these groups, standard errors of the mean are big in both groups due to high variability within groups that led to statistically no significant difference between H6 and L6. Analysis of variance showed that dose had not significant effect. Only the effect of time was significant.
- Lines 395-399. Based on these lines, why it this study different in terms of PK as detailed in lines 59-61?
As we stated in answer to comment 3, literature data on pharmacokinetic properties of ibogaine in female rats (or on comparative studies on male and female rats) are scarce (one abstract, one brief mentioning of meeting report and one research article). Our experiment was performed on Wistar rats treated per os with ibogaine; doses were 1 or 20 mg/kg; time of sacrificing 6 and 24 hours after the treatment. The difference of our experimental design and listed literature is either in strain of rats and/or applied doses and/or the way of application and/or in time points (they are reported only in Pearl et al. 1996). Thus, we decided to measure ibogaine and noribogaine concentrations in plasma at the same time points when we take samples for other measurements and observations. Our intention was not to perform PK study, but to strengthen our investigations of other parameters with both ibogaine and noribogaine measurements in the same time points as well as to enable better comparison of ibogaine effects on males and females.
- Lines 407-410. This does not seem to have a flow with the aforementioned text. There was not precedent on ROS to support these sentences
Thanks for comment. We edited this section.
- Line 516-517. this conclusion cannot be supported from the present study
Thanks for comment. You're right. We edited that sentence since we did not perform PK study. We think that now conclusion is supported from the present study.
- While a lot of data were presented the discussion focuses mostly on the antioxidant system. Some results are not even mentioned. If not needed, perhaps should not be included in this study to begin with and include only results relevant to the title and the main aim of the study.
Thanks for comment. We have extended Title, Introduction and Discussion according to presented results. Although the focus is on antioxidant system, we consider other parameters as well, but incorporated into the text in Discussion, especially when kidney and liver have been considered. Anyway, additional sentences have been added that give some explanations for observed changes.

Reviewer 2 Report
The manuscript entitled "Ibogaine has sex specific effects on the cellular antioxidant systems in rats: the study on females" presents interesting results on the effects of ibogaine on kydney, liver and erythrocytes of female rats. However, there are several similarities with references 17 and 19 from the same authors. The authors used similar methodologies, only differing the sex of the rats .
For all these reasons, and, despite the quality of the manuscript, I do not consider the results with innovation enough to be published in Life Journal.
Author Response
Dear Reviewer,
Our intention was to explore ibogaine effects in females due to general sex-based differences in pharmacokinetics, pharmacodynamics and adverse drug reactions that are well established in literature (Beierle et al. 1999; Whitley and Lindsey 2009; Soldin and Mattison 2013; Zucker and Prendergast 2020).
Besides, at the 55th Annual Scientific Meeting of The College on Problems of Drug Dependence, (Jeffcoat et al. 1994) sex-based differences in ibogaine bioavailability in rats were reported. Namely bioavailability of ibogaine was shown to be greater in females. Pearl et al. (1997) also showed sex differences in bioavailability as well as in the effects at the level of central nervous system. Furthermore, numerous fatalities associated with ibogaine use have been reported (Alper 2001; Aćimović et al. 2021) revealing a clear need for better understanding of its properties. Тhis led us to explore other potential sex-based differences in ibogaine effects. Furthermore, one of the problems with potential ibogaine application is adequate dosing and appropriate benefits without side effects that can lead to mortality. These could be different according to sex. Therefore, we tried to have not just basic scientific experimental approach but the problem we tried to solve.
In revised manuscript we referenced to some of the papers that show specific sex-based differences that are of particular relevance to some of the topics that our study deals with. Namely, glycogenolytic activity and glucose metabolism are sex-dependant (Gustavsson et al. 2010) in both rats and humans. Moreover we cited literature pointing to sex-based differences in vasomotor tone and endothelium-dependant vasodilatation that can be both sex hormone dependant and independent (Stanhewicz et al. 2018; Godo and Shimokawa 2020).
All of the above-mentioned points to the need to examine sex-based differences in the effects of ibogaine in order to better understand its mechanisms of action because these differences could lead to treatment failure or adverse effects.
In this study we showed that at the same time intervals after the application of the same oral doses of ibogaine, concentrations of both ibogaine and its primary metabolite noribogaine in blood plasma of female rats were 2-3 times higher compared to males. Histopathological changes in liver were recorded in female rats only. Moreover, they were present in all groups treated with ibogaine. Histopathological changes in kidneys 6 hours after the higher dose were also more pronounced in female rats. On the other hand, glycogenolytic activity in liver after ibogaine ingestion was less pronounced in females. Treatment with ibogaine caused AOS/redox disbalance in both male and female rats, although differences observed in liver might indicate that ibogaine had differentially activated glycogen or fatty acid dominated patterns of energetic metabolism in males versus females. An increase in catalase activity in liver of female rats but not in males indicates ROS mediated effects that can lead to oxidative damage, different H2O2 mediated signalling paths and disturbances of subtle redox regulation in a sex specific manner.
We believe that all of this should be taken into account when therapeutic usage of ibogaine is considered since sex-based differences may influence health benefit and therapeutic potential of ibogaine.
References:
- Beierle, I.; Meibohm, B.; Derendorf, H. Gender differences in pharmacokinetics and pharmacodynamics. Int J Clin Pharmacol Ther. 1999 Nov;37(11):529-47. PMID: 10584975. https://pubmed.ncbi.nlm.nih.gov/10584975/
- Whitely, H.; Lindsey, W. Sex-Based Differences in Drug Activity. Am Fam Physician. 2009 Dec 1;80(11):1254-1258. https://www.aafp.org/afp/2009/1201/p1254.html
- Soldin, O.P.; Mattison, D.R. Sex differences in pharmacokinetics and pharmacodynamics. Clin Pharmacokinet. 2009;48(3):143-57. doi: 10.2165/00003088-200948030-00001. https://www.ncbi.nlm.nih.gov/pmc/articles/PMC3644551/
- Zucker, I.; Prendergast, B.J. Sex differences in pharmacokinetics predict adverse drug reactions in women. Biol Sex Differ 11, 32 (2020). doi: 10.1186/s13293-020-00308-5. https://bsd.biomedcentral.com/articles/10.1186/s13293-020-00308-5#citeas
- Jeffcoat, R.; Cook, C.; Hill, J.; Coleman, D.; Pollack, G. Disposition of [3H] ibogaine in the rat. In NIDA Research Monograph Series; Harris, L.-S. Eds.; Publisher: National Institute on Drug Abuse, Rockville, USA, 1994; 141, pp. 309. https://archives.drugabuse.gov/sites/default/files/monograph141.pdf
- Pearl, S.M.; Hough, L.B.; Boyd, D.L.; Glick, S.D. Sex Differences in Ibogaine Antagonism of Morphine-induced Locomotor Activity and in Ibogaine Brain Levels and Metabolism. Pharmacol. Biochem. Behav. 1997, 57, 809–815. DOI: 10.1016/s0091-3057(96)00383-8. https://www.sciencedirect.com/science/article/abs/pii/S0091305796003838?via%3Dihub
- Alper, K. Ibogaine: a review. Alkaloids. Chem. Biol. 2001, 56, 1–38. DOI: 10.1016/s0099-9598(01)56005-8. https://www.sciencedirect.com/science/article/abs/pii/S0099959801560058?via%3Dihub
- Aćimović, T.; Atanasijević, T.; Denić, K.; Lukić, V.; Popović, V.; Bogdanović, M. Death due to consumption of ibogaine: case 578 report. Forensic. Sci. Med. Pathol. 2021, 17, 126‒129. DOI: 10.1007/s12024-020-00342-0. https://link.springer.com/article/10.1007%2Fs12024-020-00342-0
- Gustavsson, C.; Yassin, K.; Wahlström, E; et al. Sex-different hepatic glycogen content and glucose output in rats. BMC Biochem 11, 38 (2010). doi: 10.1186/1471-2091-11-38. https://bmcbiochem.biomedcentral.com/articles/10.1186/1471-2091-11-38
- Stanhewicz, A.E.; Wenner, M.M.; Stachenfeld, N.S. Sex differences in endothelial function important to vascular health and overall cardiovascular disease risk across the lifespan. Am J Physiol Heart Circ Physiol. 2018 Dec 1;315(6):H1569-H1588. doi: 10.1152/ajpheart.00396.2018. Epub 2018 Sep 14. PMID: 30216121; PMCID: PMC6734083. https://journals.physiology.org/doi/full/10.1152/ajpheart.00396.2018
- Godo, S.; Shimokawa, H. Gender Differences in Endothelial Function and Coronary Vasomotion Abnormalities. Gender and the Genome. January 2020. doi: 10.1177/2470289720957012. https://journals.sagepub.com/doi/full/10.1177/2470289720957012

Reviewer 3 Report
Dear Authors, these results will contribuite to better caracterized liver glycogenolytic activity and antioxidant enzymes activity in kidney and erythrocytes after a single oral dose (1 or 20 mg/kg b.w.) of ibocaine. In particular, they confirmed previous data (reference 16), infact sex differences are observed at 6 and 24 h after the treatment.
Furthermore, these results uphold the need to carry out preclinical studies in both sexes since important pharmacokinetic and pharmacodynamic differences have been shown, which often led to therapy failure.
However, it would be interesting to continue studying the "therapeutic" antioxidant effects of ibocaine in experimental models of liver oxidative stress in rodents
Author Response
Dear Reviewer,
Thanks for affordable comments. We hope that preclinical studies will be lead that way. Our intentions are quite this way.

Round 2
Reviewer 1 Report
Substantial improvements were made compared to the previous version. I still think that comment 2 on the CNS effect still does not make sence even after revising. Add a sentence to explain how Ibogaine can affect bioavailability if it affects the CNS
Comment 3, still not referenced/referred there appropriately and leaves the reader wondering.
the paper provides some information on effect of Ibogaine on female mice. It can be usefull down the road, even though a good study would be a head to head comparison, single study in male and female mice.
It would be nice to include limitations of the present study, and future work
Author Response
Dear Reviewer,
Thank you for your comprehensive review and very useful comments…
Replies to specific comments:
Comment 1: Substantial improvements were made compared to the previous version. I still think that comment 2 on the CNS effect still does not make sence even after revising. Add a sentence to explain how Ibogaine can affect bioavailability if it affects the CNS.
Reply: We think that the sentence in the text creates misunderstanding and confusion. Ibogaine cannot affect its own bioavailability, and we did not say nor wrote that. Opposite, greater bioavailability of ibogaine in females led to stronger effects on target tissue – CNS in females comparing to males (as shown by Pearl et al, 1997, referenced in the text), that’s the point. Therefore, dosage of ibogaine per os has to be different regarding to sex. Again, we edited this sentence again. Now it should be clear what we wanted to say. The previous sentence has now been split in two sentences: one about sex specific pharmacokinetics of ibogaine and the other sentence about sex specific effects of ibogaine in CNS. Sentence about effects in CNS has been moved to more appropriate place in the first paragraph. Now the whole second paragraph is about pharmacokinetics/bioavailability.
Comment 2: Comment 3, still not referenced/referred there appropriately and leaves the reader wondering.
Reply: We edited this sentence. Now it should be clear and properly referred. Our assertion (as we tried to explain in our earlier response) comes from literature search that showed that there is no publically available detailed comparative data about ibogaine PK in both sexes in rats. One is abstract without detailed presentation of results. Just scarce time points. We had no insight into levels, time points, concentrations, just percentages… Even more, Alper in his paper convinced what he had heard from Upton on NIDA Ibogaine Review Meeting in Rockville, Maryland, 1995 (referenced by Alper, 2001). We now referenced all three reference that we had about some PK of ibogaine in rats.
Comment 3: the paper provides some information on effect of Ibogaine on female mice. It can be usefull down the road, even though a good study would be a head to head comparison, single study in male and female mice.
Reply: Here we report results obtained in females. We used the same experimental design as preformed on male rats. All of the applied procedures, equipment and reagents were the same. We treated them with the same doses of ibogaine (from the same batch); sacrifice them in the same time points after the treatment; concentrations of ibogaine and noribogaine in plasma of both sexes were measured together (at the same time under the same conditions; results for each animal of both sexes are presented in the appendix of the manuscript); all the enzyme activities were measured using the same protocols, the same reagents and the same spectrophotometers; all of the histological slides were examined by the same 3 examiners, etc. All of this allowed us to make a valid comparison of the effects of ibogaine on males and females.
Comment 4: It would be nice to include limitations of the present study, and future work.
Reply: We have expanded the discussion according to your advice. Limitations of the present study, and future work were included.

Reviewer 2 Report
Although I understand the reasons of the authors to perform tests in female rats to study sex-specific effects of ibogaine, I still have reservations concerning the originality of this study. For that reason, I recommend the rejection of the manuscript.
Author Response
Dear Reviewer,
Comment: Although I understand the reasons of the authors to perform tests in female rats to study sex-specific effects of ibogaine, I still have reservations concerning the originality of this study. For that reason, I recommend the rejection of the manuscript.
Reply: As far as we know, our manuscript is the first one to report the results of histopathological analysis of liver and kidney, amount of glycogen in liver, concentrations of TBARS, thiols and protein –SH groups, activities of SOD1, SOD2, CAT, GSH-Px, GR, GST, XOD in liver, kidney and erythrocytes, etc. of female rats after the treatment with ibogaine. Furthermore, there are (only) three publications on pharmacokinetics/bioavailability of ibogaine and noribogaine in female rats: one meeting abstract (Jeffcoat et al. 1994), one brief mentioning of what was presented at the meeting (Upton 1995 as reviewed in Alper 2001) and one peer-reviewed article (Pearl et al. 1997). All of these publications differ from our study either in strain of rats and/or applied doses and/or the way of application and/or in time points (they are reported only in Pearl et al. 1997). Originality of our work comes from the fact that nobody has published results on females that really deserves scientific attention. Our results point out also that dosage of ibogaine has to be sex specific regarding its effects.
